# Pilot Study on Exhaled Breath Analysis for a Healthy Adult Population in Hawaii

**DOI:** 10.3390/molecules26123726

**Published:** 2021-06-18

**Authors:** Hunter R. Yamanaka, Cynthia Cheung, Jireh S. Mendoza, Danson J. Oliva, Kealina Elzey-Aberilla, Katelynn A. Perrault

**Affiliations:** Laboratory of Forensic and Bioanalytical Chemistry, Forensic Sciences Unit, Chaminade University of Honolulu, Honolulu, HI 96816, USA; hunter.yamanaka@student.chaminade.edu (H.R.Y.); cynthia.cheung@chaminade.edu (C.C.); jireh.mendoza@student.chaminade.edu (J.S.M.); Danson.oliva@student.chaminade.edu (D.J.O.); kealina.elzey-aberilla@student.chaminade.edu (K.E.-A.)

**Keywords:** breath profiling, comprehensive two-dimensional gas chromatography, exhaled breath, metabolites, volatile organic compounds, population in Hawaii, Oahu residents

## Abstract

Fast diagnostic results using breath analysis are an anticipated possibility for disease diagnosis or general health screenings. Tests that do not require sending specimens to medical laboratories possess capabilities to speed patient diagnosis and protect both patient and healthcare staff from unnecessary prolonged exposure. The objective of this work was to develop testing procedures on an initial healthy subject cohort in Hawaii to act as a range-finding pilot study for characterizing the baseline of exhaled breath prior to further research. Using comprehensive two-dimensional gas chromatography (GC×GC), this study analyzed exhaled breath from a healthy adult population in Hawaii to profile the range of different volatile organic compounds (VOCs) and survey Hawaii-specific differences. The most consistently reported compounds in the breath profile of individuals were acetic acid, dimethoxymethane, benzoic acid methyl ester, and *n*-hexane. In comparison to other breathprinting studies, the list of compounds discovered was representative of control cohorts. This must be considered when implementing proposed breath diagnostics in new locations with increased interpersonal variation due to diversity. Further studies on larger numbers of subjects over longer periods of time will provide additional foundational data on baseline breath VOC profiles of control populations for comparison to disease-positive cohorts.

## 1. Introduction

Rapid, non-invasive breath screening is an attractive alternative test for diseases that commonly require lengthy diagnostic procedures, such as lung infections and certain cancers [1]. In many cases, invasive procedures such as bronchoscopy, bronchoalveolar lavage or lung biopsy must be performed to obtain tissue samples for information. This is inherently challenging for certain patients as these procedures are invasive, may require sedation, are associated with significant morbidity, and in some cases even mortality. These procedures can also be extremely unpleasant, particularly for children and the elderly who must provide such samples. Alternatively, the diagnosis and/or monitoring of lung disease through breath analysis carries numerous benefits to a patient. Patients can produce breath samples in a simple manner and samples can be collected from children or patients who are unconscious [1]. Breath collection can potentially be performed quickly and with minimal equipment for in situ healthcare offices or bedside monitoring. These implications could allow more frequent monitoring and potentially more rapid response to symptoms. Breath screening would also reduce the number of patients that must undergo more invasive procedures, alleviating certain pressures and backlog in the healthcare system [2]. The concept of chemically profiling exhaled breath is not new but has certainly gained significant momentum [3]. Research has also been performed regarding the application of breathomics during the current urgency for COVID-19 diagnosis and in differentiating the disease from other respiratory infections [4].

The scientific foundation for breathprinting having potential in clinical practices varies between diseases. In many respiratory diseases, there is a shift in the cell metabolites associated with the condition that can be exploited to differentiate health and diseases in individuals, or in the case of pathogenic conditions, there is an exploitable metabolic profile of the foreign cells. The volatile organic compounds (VOCs) detected in the breath have been shown to vary between healthy individuals and those affected by conditions such as chronic obstructive pulmonary disorder (COPD), diabetes, and lung cancer [5,6]. These conditions represent significant health disparities in Hawaii and effective breathprinting tools could significantly contribute to the redressing of these discrepancies in health care, reducing considerable backlog in the medical system, and improving patient care. VOC targets in exhaled breath are largely comprised of a wide range of compounds such as hydrocarbons, alcohols, aldehydes, ketones, volatile fatty acids, and sulfur-containing compounds [7].

While the idea of obtaining a breathprint to diagnose lung disease has been proposed in the past, several obstacles have prevented the rapid development of commercial biomarker sensors. Commercial sensors are well-developed [8], yet they must be tuned for specific marker compounds at a known concentration range in order to provide accurate and reliable results. The analytical identification of disease breath biomarkers for the purposes of creating such sensors is, however, challenging due to their complex mixture and wide concentration range. Even in healthy individuals, little is known about the breath profile, which is often a major inherent obstacle preventing disease biomarker identification for low-cost bedside sensor development. Sex has been shown to affect the exhaled breath profile [9], yet other factors that affect this profile remain largely uninvestigated. For example, emotional state is known to impact levels of chemicals emitted into the air from exhaled breath [10]. Understanding the control population used in studies is crucial for advancing work in the area of exhaled breath analysis. This is even more important in regions where there is high population diversity, which could introduce variation in breath profiles within a healthy population. Without a fundamental understanding of exhaled VOCs across a wide range of populations, there will be challenges with the realistic implementation of exhaled breath diagnostics on a global scale. 

One major ongoing shift in exhaled breath diagnostics is the introduction of comprehensive two-dimensional gas chromatography (GC×GC) as an analytical tool for research studies. The benefit of using GC×GC for exhaled breath diagnosis is that it increases peak capacity beyond that of traditional one-dimensional gas chromatography (GC) and therefore allows improved performance in comprehensively characterizing a sample. In exhaled breath studies, GC×GC has been used to better understand disease biomarkers for asthma, lung disease, and tuberculosis [11,12,13] among others. These studies tend to be held in centralized locations with very different populations than those in Hawaii, and therefore, a key question is whether these complex VOC profiles and results of breath diagnostic research can be applied amongst more diverse populations outside of where the control groups were originally assessed.

The objective of this research was to establish a breathprint sampling method and conduct a pilot range-finding study to investigate the variance of breath compounds in a Hawaiian population using gas chromatographic techniques. The number of subjects was kept intentionally small for this first pilot study in an attempt to look at intra-individual differences over time, from data collection on three separate visits for each individual. The population used in the study were healthy adults that met strict criteria for inclusion in the study, as further elaborated in the methods section. In particular, there was an interest in examining the variation in the population in Hawaii as a premise for future large-scale studies. Because of high diversity, immigration, and tourism, the population in Hawaii may have high variance concerning the exhaled breath profile from healthy individuals. As GC×GC is currently emerging as a valuable tool for the complexity of exhaled breath samples, this was the instrumentation chosen for the analysis. On the instrument used for this particular study, GC×GC was able to be combined with dual detection using a flame ionization detector (FID) and quadrupole mass spectrometer (qMS). The dual detection approach enables molecule discovery and identification using the qMS detector and more accurate linear quantification using the FID. Prior studies outline the benefits and data workflows for GC×GC-qMS/FID for VOC analysis [14,15].

## 2. Results

### 2.1. Pre-Trial Tube Selection

In order to distinguish between breath samples collected from human subjects in Hawaii which has not been done before, the most appropriate type of sorbent tube suitable for breath collection was first determined using gas chromatography – mass spectrometry (GC-MS). The three types of sorbent tubes analyzed were: Tenax TA, Biomonitoring, and Odour/Sulfur. The Tenax TA tubes are a general-purpose sorbent tube that covers an analyte range from C_6_–C_30_. The Biomonitoring tubes are Tenax TA tubes with the addition of graphitized carbon, and cover an analyte range from C_4_–C_20_. The graphitized carbon is meant to extend the range of lighter volatiles detected. Comparing the literature [16,17,18], there seem to be minimal differences between the Tenax TA and Biomonitoring sorbent tubes. The Odour/Sulfur tubes are Tenax TA tubes with the addition of SulfiCarb sorbent, recommended by the manufacturer for monitoring a wide range of compounds and reactive sulfur species, and cover an analyte range from C_6_–C_30_ [19]. Therefore, the Odour/Sulfur tubes should ideally improve sulfur compound recovery. These sorbent tubes were initially chosen in this study because they are comprised of specific sorbents to target breath VOCs, and likely to perform well on generating breath profiles.

To test the different types of sorbent tubes, a breath volatiles reference mix (VRM) was injected onto the tubes as the first point of comparison, and samples from human subjects were collected onto the tubes as the second point of comparison. The VRM injections on the tubes allowed for assessment that desorption parameters and tube conditioning approaches were appropriate, while the samples from human subjects helped to better understand how real samples, impacted by factors such as moisture and breakthrough volume, would perform. The real breath samples allowed assessment of whether compounds within the breath were visible and whether differences could be detected using the method, therefore allowing confidence in the volume of sampling before proceeding. Figure 1 illustrates comparable overall peak areas of the VRM compounds detected in each type of sorbent tube. There were no major differences between tubes using the VRM alone. Although not statistically significant, the Tenax TA tubes showed reduced variability compared to the Biomonitoring and Odour/Sulfur tubes. 

In addition to spiking the three different sorbent tubes with a reference mix, breath samples from a preliminary cohort of individuals (n = 3) were also utilized to compare tubes. The results demonstrated that Odor/sulfur tubes did not recover as many lighter volatiles, as highlighted by the green boxes in Figure 2. 

Overall, Biomonitoring and Tenax TA tubes performed similarly to one another, as can be further noted in Appendix A. While the Odour/Sulphur tubes should ideally improve sulfur compound recovery [19], the other two sorbent types showed an improved coverage of a wider range of analytes. Odour/Sulpur tubes consistently showed a lower recovery of compounds across all three subjects, as noted in Figure 2 and Appendix A. The differences between Tenax TA and Biomonitoring tubes were more subtle from subject to subject (Appendix A), and may have also been attributed to differences in intra-individual breaths on different sorbent types rather than the sorbent itself. The benefit of graphitized carbon in the Biomonitoring tubes meant to extend the range of lighter volatiles detected was not observed, based on the data shown above of the different tube comparisons. With the results obtained from the VRM standard and the human subject tube data, Tenax TA was identified as the more compatible sorbent tube for the following human breath studies conducted. Although not statistically significant due to high variability in the Biomonitoring tubes, Tenax TA generally appeared to have a higher abundance of compounds for each individual (Figure 2), as well as the lowest sampling variability (Figure 1). These independent findings also align with a recent study that investigated six common types of sorbent tubes. This investigation, which used nearly identical desorption parameters as the current study, concluded Tenax TA tubes recovered the widest range of analytes and best reproducibility, among other benefits [20]. It should also be noted that the range of analytes recovered in the pre-trial tube selection study was comparable with Section 2.2. Variability of these compounds from the pre-trial tube selection was further improved upon in the full trial due to the increase in resolution between individual peaks and the removal of artifacts from compounds of interest.

### 2.2. Post-Trial Compound Identification

The chromatographic separation obtained in the human subjects trial is demonstrated with chemical standards in Figure 3. Each human subject had only a subset of these compounds and therefore the mix of compounds is demonstrated cumulatively with standards in the depiction below. In this figure, compounds are well resolved from one another and use the majority of the contour plot space. One must note that from sample to sample, the use of this space varied due to the different compositions of each sample. However, an apex plot is shown for all compounds identified across the study, and it can be seen that the cumulative presence of all potential compounds in breath samples benefited greatly from the increased peak capacity of GC×GC. All compounds existing in the same vertical plane would not necessarily be possible to resolve in the 1D GC analysis. The GC×GC-qMS data were used predominantly for peak identification and the GC×GC-FID data were used for quantitative information. Previous studies describe the use of this dual detection technique in combination with GC×GC and the full workflow used for processing [14,15].

All qMS data from every sample were combined to generate a list of over 100 components identified in the trial. The FID data were then processed and the number of components identified as peaks on average for each subject is shown in Table 1. Based on the presence of peaks in the FID and the matching MS identification, the total number of compounds that could be tentatively identified are shown in Table 2. Furthermore, 22 compounds were found to surpass the F_crit_ value in their respected sample groups (see Table 2). This demonstrates that although the breath samples themselves started off as being quite complex, with a dedicated workflow to eliminate noise and other interferences, the resulting number of compounds with high variation was relatively low as listed in Table 1 under total number of compounds different than room air. It is important to note that if all samples were combined together, the number of peaks physically present in the combined samples would far surpass the peak capacity of a one-dimensional technique. However, Table 1 outlines the attempt to focus on compounds based on variance rather than based on presence within the sample. The list of components actually identified in Table 1 represents compounds that were (1) not representative of analytical artifacts, (2) present at levels that were variable (e.g., up- or down-regulated in the samples), and (3) had compound identifications that were reliable enough to report compound identity based on standard injection and retention information. Upon collection of samples, breath samples and control samples were gathered directly after one another to reduce the sample composition variability. Longer durations between samples would potentially have been problematic in terms of room composition shifting due to room air circulation.

Although, ultimately, the number of compounds of focus per subject was few, the compounds of importance were different from subject to subject, demonstrating that an analytical technique that can theoretically resolve every possible compound of importance from the potential hundreds of compounds that can appear in a contour plot is beneficial. The compounds of importance identified in the feature reduction process would not necessarily have been the same compounds highlighted in a one-dimensional GC approach that does not sufficiently resolve all possible components for further processing and characterization. Since the variation between days for an individual can be small, and the variation between subjects can be large, an analytical technique that provides superior resolution is highly beneficial. Since this type of research is non-targeted, exploring all possible compounds that can be detected, the high-capacity nature of GC×GC provides the opportunity to obtain the highest quality return on molecular differences between samples.

In order to highlight common components between the various breath samples, based on the analytes reported in Table 2 all breath samples were reprocessed to check if some compounds of a specific individual were present in the other subjects’, potentially improving specificity. The goal of this stage of the analysis was to incorporate an assessment of intraindividual variation. All reprocessed data were cross compared to determine the subject-specific and recurrent analytes. It appeared that there were a select few analytes present in every healthy breath sample, while some compounds showed to be specific to the healthy human subject it was acquired from. The compounds identified consistently in the subject population (100%), including consistently on each day of analysis per subject with low intraindividual variation (see Table 3) were: acetic acid, dimethoxymethane, n-hexane, and benzoic acid methyl ester. Compounds found across 6 of 7 subjects (86%) also included benzene and benzaldehyde. Compounds found across 5 of 7 subjects (71%) also included benzofuran and acetophenone. It is important to note that the colors in Table 3 represent an assessment of variation for each subject. The table represents all compounds detected labelled as significant in the study, however, for some compounds, the intraindividual variation was high from visit to visit and this is captured by the yellow squares within the table. The group of components towards the top of the table, with mostly green and yellow squares, may be interesting markers to monitor in larger-scale studies moving forward with additional research as their variation appears to be lower from an inter- and intra-individual perspective. Compounds that fluctuate in terms of their presence and absence, or appear to have increased variability would be less valuable as biomarkers as they may not have a stable baseline to refer to within the population.

## 3. Discussion

The compounds that appear lower in Table 3 appear to have higher intraindividual variation within the subjects monitored within this study. Increasing subject participants would help to improve the understanding of the extent of this variation. However, these compounds did not appear to be very stable in abundance in individuals from this small cohort studied. This would assert that caution should be used if attempting to use these particular compounds as disease markers when moving a research study conducted in one locale to another area. It suggests that an evaluation of healthy individuals within other regions, including Hawaii, may be necessary to understand whether biomarkers of disease developed in one region of the world can be realistically applied in another area. If these particular compounds were being monitored for upregulation or downregulation in disease diagnosis, and that concept is applied in a new region where there is naturally a higher variability in those compounds within healthy individuals, it could raise the risk of false positives in disease diagnosis. The data provided in this study allowed a starting assessment of the variance of a healthy population in Hawaii to be characterized as a pilot trial. Further data are needed with a larger subject cohort to make conclusions about certain breath markers being stable for use as disease markers within this population. 

Acetic acid was a compound that was identified in every patient within this study and was found to be significantly different from room air using Fisher Ratio variance analysis. Acetic acid is a known marker monitored for gastrointestinal reflux disease (GERD) as well as for monitoring cystic fibrosis patients for potential lung infections [21]. While the increase in this particular breath marker is important in indicating the difference between a healthy subject, a subject with GERD, and a subject with a lung infection, it has also been noted that the accurate quantification of this biomarker is essential in order to ensure that appropriate classification of health status can be achieved. This is largely because acetic acid is present in low levels (ppbv) in exhaled breath from subjects in many studies, and therefore knowing the background levels is important if applying this type of research to new populations of individuals [22]. Acetic acid has been demonstrated as being taken up by human primary tracheobronchial epithelial (TBE) cell lines, as well as lung adenocarcinoma cell lines (A549, Lu7466), while being released by human epithelial cervical carcinoma (HeLa) cell lines [23].

The compound dimethoxymethane (also known as methylal) was also consistently identified in breath samples within this study. Dimethoxymethane is a known breath compound in exhaled breath [7]; however, it has recently been demonstrated that females release significantly lower amounts of dimethoxymethane in their breath samples compared to male subjects [9]. It is possible that other genetic factors could contribute to differences in dimethoxymethane production in exhaled breath as well, though to our knowledge this has not yet been investigated. Interestingly, dimethoxymethane has only been identified in breath from human subjects and not in other bodily secretions [7].

*n*-hexane is a compound found in all matrices collected from healthy human subjects including feces, urine, breath, skin, milk, blood, and saliva [7]. *n*-hexane is known to be released by lung cancer cells (NCI-H2087) at a much higher level than found in baseline levels of healthy subjects. However, very few reports exist on what the actual baseline level of hexane is found to be in subjects, as studies most often report the results of significant difference between a control group and the group of samples with elevated levels. Therefore, further understanding of absolute concentration and variance amongst individuals within different populations would assist in ensuring that a test using this biomarker remains effective when deployed to subjects outside of a strict study control group. The importance of accurate quantification in breath studies for this particular reason has been highlighted previously [22].

Benzoic acid methyl ester was consistently identified across subjects. Esters of benzoic acid are commonly found on the skin of healthy individuals [7] including benzoic acid dodecyl ester, benzoic acid tridecyl ester, and benzoic acid tetradecyl ester [7]. These are generally larger molecules that would be less volatile than benzoic acid methyl ester. Benzoic acid methyl ester is not a commonly cited VOC in exhaled breath [7]. This compound may potentially be related to the subjects in this particular study, perhaps linked with factors such as environmental exposure. This compound is found as a floral aroma in many plants.

Benzaldehyde, benzene, benzofuran and acetophenone were also identified in a large majority of subjects (>70%). Benzaldehyde and benzene have been commonly identified in all matrices collected from healthy human subjects, including feces, urine, breath, skin, milk, blood and saliva [23]. This is also true of acetophenone with the exception of milk [23]. Benzofuran has been reported as a chemical in exhaled breath, skin, and milk from healthy subjects [7]. It should also be noted that two prominent breath VOCs, acetone and isoprene, were not detected within this study. The sorbent-based collection methods used in this study may have impeded the ability to collect, focus, and inject these compounds onto the instrument and therefore may have contributed to their lack of detection. It is also possible that these compounds fell below the limit of detection within this study or that they were not present in the sample. This should be a focus of further investigation when moving towards studies incorporating a larger number of individuals.

Additionally, the authors note that specific absolute concentrations of compounds were not calculated in this study, and may be beneficial to include in future studies, especially for the core breath profile compounds that are detected. The current study focused on relative quantities of compounds to one another and on variance analysis rather than on calibration and performing absolute quantification. In relating this data to different diseases or disorders within the population, absolute quantification may be a more robust approach.

This is the first time an exhaled breath study has focused on profiling healthy subjects in Hawaii. Although the data only represent 7 subjects and therefore has limited ability to make broad inferences on the population, it is important to represent different and diverse populations to understand the implications of using exhaled breath as a diagnostic or health monitoring procedure in the future. Understanding the baseline breath profile of subjects across different populations will assist in developing breath tests that are more accurate and reliable and reduce the possibility of false negatives or false positives when tests can eventually be implemented. This work is a pilot study that assists in identifying consistencies and differences in a small group of individuals, and therefore has limited statistical significance compared to a study incorporating more individuals and data. The next phase of this work would involve scaling up the study to include a large number of individuals and tracking information like ancestry alongside the data for further clarity.

## 4. Materials and Methods

### 4.1. Human Subjects

Healthy volunteers were recruited from within the University community according to approved Institutional Review Board (IRB) procedures under IRB Protocol # CUH052. Individuals self-reported their qualifications for the study based on inclusion criteria and exclusion criteria. Inclusion criteria were defined for this initial cohort as lifetime non-smokers, male or female, adults (age 18–54), and within normal body mass index (BMI) range (18.5–25). Choosing criteria for BMI and smoking activity level served to reduce variation in metabolic rates between individuals in this initial cohort.

Additionally, exclusion criteria included the previous history of neonatal lung development complications/conditions, previous conditions that required the use of a medical ventilator, cold, flu, or respiratory tract infection symptoms exhibited at the time of breath sample collection or in the past two weeks. Vulnerable populations were also excluded from the study. After determining study qualifications based on the criteria, informed consent was obtained according to Protocol #CUH052.

### 4.2. Pre-Trial Tube Selection

Investigating literature on breath analysis and on manufacturer’s recommendations, there were three sorbent tubes that would be suitable for breath collection: Tenax TA, Biomonitoring, and Odour/Sulphur (Markes International Ltd., Llantrisant, UK.). In order to determine which of the three tubes to use in-house for this study, a pre-trial study was conducted on a small range of samples to confirm a suitable choice. Two approaches were used for this confirmation. First, three of each sorbent tube were used with chemical standards that were representative of exhaled breath. A 10 ppm breath VOC standard was prepared by mixing two different custom mixes, one commercial mixed standard, and several individual standards. Custom Mix 1 contained 2-ethyl-1-hexanol, 1-propanol, 2-propanol, 2-butanone, cyclohexane, and 2-methylfuran in P&T methanol/water (GC Grade, Restek Corporation, Bellefonte, PA, USA). Custom Mix 2 contained styrene, 2-methylpentane, 3-methylpentane, 2,4-dimethylheptane, 2-methylhexane, naphthalene, and 1,2,3-trimethylbenzene in P&T methanol (GC grade, Restek Corporation). Commercial mix 1 contained benzene, toluene, ethylbenzene, m-xylene, o-xylene, and p-xylene in P&T methanol (certified reference mixture, Restek Corporation). Individual standards for hexanal, heptanal, and dimethyl trisulfide (DMTS) were also used (analytical standard grade, Sigma-Aldrich, St. Louis, MO, USA). These mixes were all combined at a concentration of 10 ppm to create the breath VOC mix. For each of the sorbent tubes, 1 µL of the breath VOC mix was injected onto the sorbent tube, with 1 µL of a 10 ppm saturated alkanes mix (C_7_–C_30_ in hexane, certified reference material grade, Supelco, Bellefonte, PA, USA), and 1 µL of 10 ppm d_5_-chlorobenzene (GC grade, Restek Corporation). Second, three human subjects contributed breath samples on one of each tube. Breath samples were collected as described below under the same IRB Protocol. Following the pre-trial tube selection, the Tenax TA sorbent tubes were used for all collected samples.

### 4.3. Breath Sample Collection

Each volunteer was advised to avoid consuming any food or beverages other than water at least 2 h before sample collection. Each participant was required to sit in place for at least 10 min in an isolated room, and to complete a lifestyle questionnaire prior to breath sample collection. The lifestyle questionnaire included questions about the participants’ activities in the preceding week (including recent food intake, sleep, activity level, exercise, prescription medication and alcohol consumption). This was repeated on three visits with each study participant, with a minimum of 48 h in-between visits.

All sample collection was completed within a 3-month period (precisely between early July and late October 2019). Participants submitted breath samples by breathing at a regular rate into a Bio-VOC sampler (Markes International Ltd.) following the recommended manufacturer’s procedures. Subjects were asked to exhale a single slow vital capacity breath into the Bio-VOC. The collected air was immediately transferred to a ¼ × 3½ stainless steel conditioned Tenax TA thermal desorption (TD) tube (Markes International Ltd.) following the direction of sampling to capture VOCs present in the trapped air. This process was repeated two more times in order to collect three single breaths as one sample, resulting in 525 mL of breath collected onto a single tube. For each set of breath samples collected from subjects, an equivalent sample of room air was collected at the same time as the subject’s visit. Room air was collected into the bio-VOC sampler by pumping the handle and expelling the room air onto a sorbent tube. The number of room air collections performed was matched to the breath sample, meaning that 3 full bio-VOC samplers of room air were expelled onto a single sorbent tube. These control samples were used to eliminate adsorbent artifacts and to characterize background compounds to reduce the reporting of compounds that were not relevant to the exhaled breath samples.

### 4.4. Pilot Trial of 7 Subjects (105 Subject Samples)

Additionally, a cohort of individuals was recruited to provide breath samples for the pilot study. A total of 7 subjects were used for this part of the research study. Each individual provided samples in three separate sessions. During each visit, the individual provided 5 breath samples, each including 3 individual breaths. This provided 15 sorbent tubes from each individual. Visits were a minimum of 48 h apart. Prior to breath collection, each TD tube was reconditioned for 30 min at 300 °C with a flow of ultra-high purity nitrogen (Airgas, Radnor, PA, USA) with a pressure of 20 psi, which equated to a flow rate of 60 mL/min. Tube reconditioning was always performed offline from the system on a TC-20 instrument (Markes International Ltd.) using a flow of ultra-high purity nitrogen (Airgas). Tubes were stored at room temperature with brass long-term storage caps in an airtight screw-capped container before breath collection. TD tubes with samples were coded (using tube number and date of data acquisition as DD/MM/YYYY) to minimize bias during data acquisition. The TD tubes were stored at 4 °C until data acquisition using a Unity-xr thermal desorption (TD) system (Markes International Ltd.) for sample introduction and comprehensive two-dimensional gas chromatography and dual-channel detection with quadrupole mass spectrometry and flame ionization (TD-GC×GC-qMS/FID) for analysis.

### 4.5. GC-MS Analysis

The pre-trial tube selection was performed on a one-dimensional gas chromatography system with quadrupole mass spectrometry (GC-MS). Sample analysis was carried out utilizing a Unity 2 series thermal desorber (Markes International Ltd.) and conducted on a Focus GC coupled with a Dual Stage Quadrupole II (DSQ II) Mass Selective Detector (MSD) (Thermo Scientific, Bellefonte, PA, USA).

Prior to desorption, a leak test was performed followed by a 1 min prepurge with a trap flow of 50 mL/min. Primary desorption of the sample took place at 300 °C for 5 min with a trap flow of 50 mL/min and a split-flow of 20 mL/min. The sample was recondensed at 10 °C on a general-purpose carbon cold trap for C_4/5_ to C_30/32_ (Markes International Ltd.). The cold trap was purged for 1 min with a flow of 50 mL/min, then heated at the maximum heating rate to 320 °C for 3 min. Following desorption, all tubes were reconditioned offline for 30 min at 330 °C on the TC-20 at 20 psi, and capped with brass long-term storage caps (Markes International) to prepare for re-use. The standby split-flow on the thermal desorber was set at 10 mL/min, and the flow path temperature at 150 °C. The GC cycle time was set for 30 min and the minimum carrier pressure at 5 psi. Analyte separation was accomplished using an Rxi-624Sil MS capillary column (Restek Corporation, 30 m × 0.25 mm ID × 1.4 µm film thickness) using ultra-high-purity helium as the carrier gas at a constant flow rate of 1.0 mL/min (Airgas). The GC oven program started at 35 °C where it was held for 5 min, followed by a temperature increase of 5 °C/min up to 240 °C, which was maintained for 5 min. The MS transfer line and ion source were set to 250 and 200 °C, respectively, and the MSD was operated in full electron ionization (EI) scan mode from 45 to 450 *m/z* at a scan rate of 5 scans/s.

### 4.6. GC×GC-qMS/FID Analysis

Sample analysis for the human subjects trial was conducted with a Thermo Scientific Trace 1300 gas chromatograph coupled to an ISQ 7000 single quadrupole mass spectrometer and Trace 1300 flame ionization detector (FID). The column junction was equipped with a reverse fill/flush (RFF) INSIGHT flow modulator (SepSolve Analytical Ltd., Peterborough, UK). An Rxi-624Sil MS column (30 m × 0.25 mm ID × 1.4 μm film thickness, Restek Corporation) was used in the first dimension. A Stabilwax column (5 m × 0.25 mm ID × 0.25 μm film thickness, Restek Corporation) was used in the second dimension. The flow rate in the first dimension column was 1.00 mL/min, the auxiliary gas flow was 20.00 mL/min, and the resulting calculated flow rate in the bleed line (5 m × 0.1 mm ID) was 1.00 mL/min. The loop dimensions were 0.53 mm ID × 1133 mm, resulting in a loop volume of 25 μL. The modulation period (PM) was 2.5 s, and the flush time was 100 ms, which was held constant throughout the full duration of the run. The calculated flow rate in the second dimension column was 17.9 mL/min. The carrier gas was ultra-high purity helium (Airgas). The flow was split with a ratio of 4.5:1 between the FID and qMS. The GC oven started at an initial temperature of 60 °C, held for 3 min, was increased to a final temperature 250 °C at a rate of 5 °C/min, and held for 5 min, resulting in a total run time of 46 min. The transfer line and the ion source temperature were held at 280 °C. The qMS, characterized by a maximum scan speed of 20,000 amu/s, was operated in electron ionization (EI) scan mode with a resulting data acquisition rate of ~41.5 Hz for the mass range of 40–300 *m/z*. The FID was set to 250 °C and was operated with an acquisition rate of 120 Hz. The flow rate for hydrogen (ultra-high purity, Airgas) was 35.0 mL/min. The flow rate for air (ultra-zero purity, Airgas) was 350 mL/min. The flow rate for the nitrogen makeup gas (ultra-high purity, Airgas) was 40 mL/min.

TD tubes containing participants’ breath samples were brought from 4 °C to ambient temperature (~22 °C) for at least 5–10 min, before analysis. Each tube was injected with 1 µL of 10 ppm d_5_-chlorobenzene (GC Grade, Restek Corporation) in HPLC grade methanol (GC Grade, Restek Corporation) using a micropipette. Helium was directed to the Unity-xr using a dynamic headspace adaptor and inert tubing and redirected through the roof of the GC oven in an insulated transfer line containing uncoated fused silica. Desorption of each sample in the TD tubes was carried out in the Unity-xr, which underwent two-step desorption; primary desorption of the sample took place with a trap flow of 50 mL/min and split flow of 20 mL/min at 300 °C for 5 min following a 1 min nitrogen dry purge. The sample was re-condensed at −10 °C on a general-purpose carbon cold trap for C_4/5_ to C_30/32_ (Markes International Ltd.). The cold trap was then rapidly heated for secondary desorption at 320 °C for 3 min following another 1 min nitrogen dry purge. Thermal desorption was controlled using Chromspace (v. 1.5.1.1, SepSolve Analytical Ltd.). The acquisition was controlled through Chromeleon software (version 7.2.9, Thermo Scientific, Waltham, MA, USA). 

Data acquisition was performed for both datasets using Thermo Scientific Chromeleon V.7.2.9. GC-qMS data were processed with the same software. GC×GC-qMS *.raw files were exported, converted into *.cdf format, and imported into ChromSpace software V.1.5.1.1 (SepSolve Analytical Ltd.) for processing. GC×GC-FID files were exported as *.cdf and imported into ChromSpace software V.1.5.1.1 (SepSolve Analytical Ltd.) for processing. Data acquisition of collected samples was completed within 2 weeks of sample collection.

### 4.7. Data Treatment

GC-MS data were treated with the following procedure. MS detection was performed using the ICIS detection algorithm with an area noise factor of 5, a peak noise factor of 150, and a baseline window of 100. The noise method was repetitive. The minimum peak width was 3, multiplet resolution was 10, area tail extension was 5, and area scan window was 0. Peak widths were not constrained. Peak-dependent correction was used with a left region bunch width of 3 spectra, peak spectrum bunch width of 3 spectra, and right region bunch width of 3 spectra. Mass spectra were searched to the mainlib and replib from the National Institute of Standards and Technology (NIST) 2017 MS library. Components from the samples were added to a component table and MS quantitation peaks, confirming peaks (2) and peak detection parameters were manually verified for each compound. Aligned peak reports were exported as *.csv files for analysis in Microsoft Excel.

GC×GC-qMS data were treated with the following procedure. Baseline correction and peak detection of all acquired data files were carried out using Chromspace software (SepSolve Analytical). Dynamic baseline correction was performed on imported *.cdf files with a peak width of 0.4 s. Stencils for the peaks of interest were obtained by applying the curve-fitting algorithm for peak integration with a 3-point Gaussian smoothing function. The minimum peak area was 0.0, the minimum peak height was 600,000, and the minimum peak width was 0.000. Parameters for peak merging included a tolerance of 2%, overlap of 2%, intensity of 2%, and correlation of 0.5. Subpeak apex windows for fronting and tailing were set to 2% for both low and high PM. Compound identifications were supported by searching the National Institute of Standards and Technology (NIST) 2017 library, in combination with retention time matching with breath VOC standards when possible. The standards that were available to confirm compound identities included dimethyl trisulfide, dimethyl disulfide (DMDS), hexanal, heptanal, heptane, octane, nonane, decane, dodecane, tridecane, pentadecane, hexadecane, heptadecane, octadecane, nonadecane, eicosane, heneicosane, tricosane, tetracosane, pentacosane, hexacosane, heptacosane, octacosane, nonacosane, triacontane, benzene, ethylbenzene, toluene, m-xylene, o-xylene, p-xylene (inject 1 microliter in td tube with the run). GC×GC-qMS data were used to generate stencil patterns overtop peaks of interest for each subject after the stencil identification was performed as above.

GC×GC-FID data were treated with the following procedure. Top Hat baseline correction was used on imported *.cdf files using a peak width of 0.4 s. Stencils obtained from GC×GC-qMS data processing method were applied to FID files, and the stencil was transformed manually to align over FID peaks. Peak detection was performed using the local regions of interest produced by these stencils with a minimum peak height of 0.0, minimum peak height was 0.0, and minimum peak width was 0.000. This allowed for the detection of all peak areas within the stencil region. The peak height was used to calculate calibration curves. Parameters for peak merging included tolerance of 10%, overlap of 10%, intensity of 0.5%, correlation of 0.0%. Stencils were adjusted manually to ensure consistent integration for all concentration levels. 

Due to the limited scan rate of the qMS detector, the approach on this type of instrumentation is to use the dual-detection system to obtain the best information from each detector. The qMS data stream is used to generate stencils where every peak is identified based on the processed mass spectrum and library search, in combination with a comparison of retention time and standard injection data. The stencils are then applied over the FID data to identify peaks in this single-channel detector. The peak integration is performed at this stage on the FID data where the acquisition rate is much higher and provides sharp and accurate peak shapes. This approach is outlined in previous publications including detector acquisition rates, number of acquisitions across a peak, and data processing workflows [14,15] and therefore are not described in detail here. 

Exported peak areas were used to calculate Fisher Ratio for each compound for each subject comparison, treating day and control as individual groups (e.g., 1 subject’s samples and controls would generate 6 groups). All breath samples for each subject were grouped separately by the day they gave breath, and control samples from each day were grouped as one group. Fisher Ratio was calculated according to the formula below (equation 1). Fisher Ratios was utilized to discriminate control samples from subjects’ samples, thus Fisher Ratios that exceeded the critical F value (F_crit_) were considered to be significant. The F_crit_ value is determined using the number of groups, number of samples within each group, and the significance level (α = 0.05). The purpose of using the Fisher Ratio to select features of importance is to assess the variance of the chemical markers across different groups of samples.
(1)$$F=\frac{between-group\;variability}{within-group\;variability}$$
(2)$$\mathrm{Where}\;between-group\;variability=\sum_{i=1}^{K}\frac{n_{i}{\left({\bar{Y}}_{i}-\bar{Y}\right)}^{2}}{K-1}$$
(3)$$\mathrm{And}\;within-group\;variability=\sum_{i=1}^{K}\sum_{j=1}^{n_{i}}\frac{{\left(Y_{ij}-{\bar{Y}}_{i}\right)}^{2}}{\left(N-K\right)}$$

## Figures and Tables

**Figure 1 molecules-26-03726-f001:**
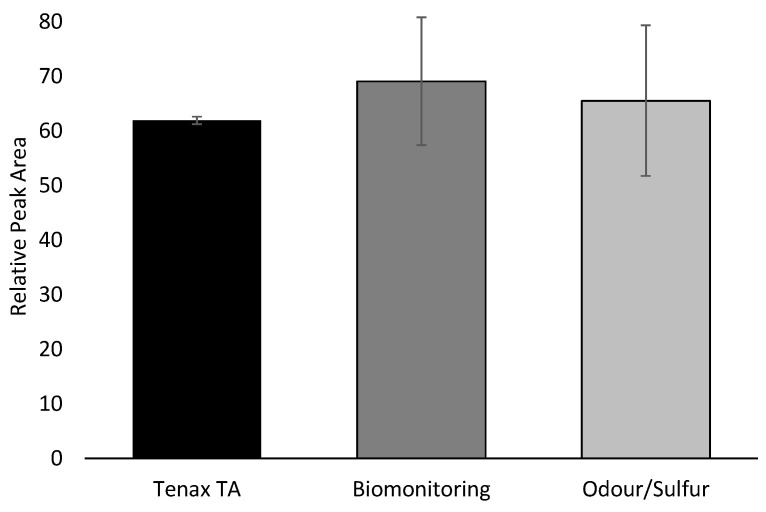
Sum of relative peak area for all normalized compounds of interest obtained using gas chromatography—mass spectrometry (GC-MS) for each sorbent tube type. Comparable total peak areas of the VRM compounds detected in each type of sorbent tube tested; no significant difference observed.

**Figure 2 molecules-26-03726-f002:**
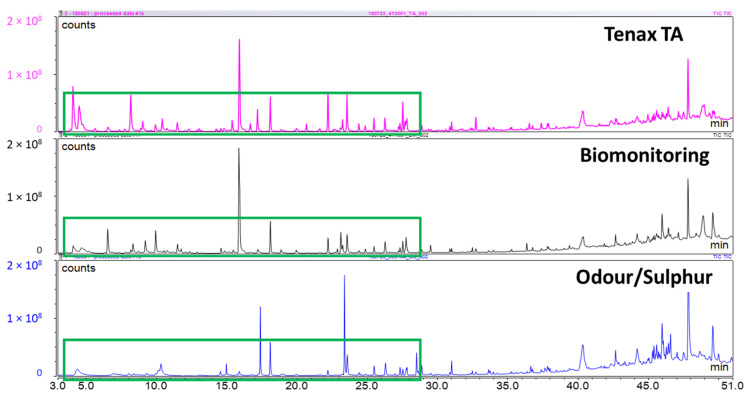
Total ion current gas chromatograms of Subject 2 on each tube type. Differences from sorbent to sorbent could have been attributed in part to differences in breath samples provided consecutively. Results for Subject 1 and 3 are available in Appendix A.

**Figure 3 molecules-26-03726-f003:**
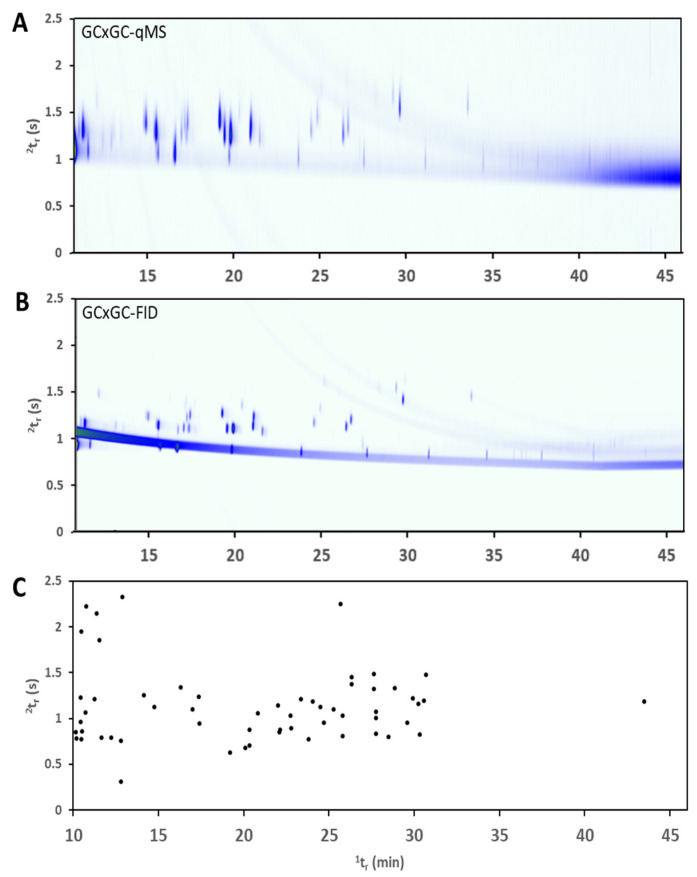
Plots illustrating comprehensive two-dimensional gas chromatographic separation. (**A**) Contour plot of standards generated with quadrupole mass spectrometry detection. (**B**) Contour plot of standards generated with FID detection. (**C**) Apex plot of all breath compounds identified from human subjects with artifacts removed.

**Table 1 molecules-26-03726-t001:** Total number of components found in breath samples for each subject using comprehensive two-dimensional gas chromatography–flame ionization detection (GC×GC-FID).

Subject Code	Total Number of Components Found (FID)	Total Number of Compounds Identified	Total Number of Compounds Different than Room Air
02	22	21	4
03	21	19	5
06	18	16	1
08	34	24	6
09	26	21	6
10	137	36	4
11	22	16	4

**Table 2 molecules-26-03726-t002:** Compounds identified using comprehensive two-dimensional gas chromatography–quadrupole mass spectrometry/flame ionization detection (GC×GC-qMS/FID). Tentative analyte identifications were made using qMS data and reported retention times are based on FID data. (Note: ^1^t_R_ = first dimension retention time and ^2^t_R_ = second dimension retention time).

Molecules	CAS #	^1^t_R_ (min)	^2^t_R_ (s)	Compound Identified in Subject(s)
iodomethane	74-88-4	6.7292	0.8074	08, 10, 11
2-aziridinylethyl	4025-37-0	6.7618	0.4992	08, 09
(1R,2R)-2-amino-1-phenylpropan-1-ol	492-39-7	6.7693	0.8805	11
carbonyldiamide	57-13-6	7.3336	1.3717	02, 08
3-methyl-2-butanamine	598-74-3	7.4152	1.4788	02, 11
nitrous oxide	10024-97-2	7.4389	1.5156	06, 09
3-methylpentane	107-83-5	9.0882	0.8508	10
2,4-dimethylhexane	589-43-5	10.1532	0.8506	11
*n*-hexane	110-54-3	10.1837	0.7824	02, 03, 06, 08, 09, 10, 11
dimethoxymethane	109-87-5	10.4252	0.9655	02, 03, 06, 08, 09, 10, 11
tetramethyloxirane	5076-20-0	10.4467	1.2284	10
propanedioic acid	141-82-2	10.4649	1.9485	08, 11
methylcyclopentane	96-37-7	10.4809	0.7724	08, 10, 11
1,2,4-trifluorobenzene	367-23-7	10.5108	0.8571	9
methanesulfonic	75-75-2	10.7328	1.0696	02, 03, 08, 09
(Methylsulfinyl)(methylthio)methane	33577-16-1	10.758	2.222	11
methylene chloride	75-09-2	11.2607	1.2145	08, 10, 11
2-butanol	78-92-2	11.365	2.1428	10
3,3,4,4-tetrafluorohexane	110-54-3	11.5197	1.8566	11
cyclohexane	110-82-7	11.6421	0.7868	10
benzene	71-43-2	12.2093	0.7881	02, 03, 06, 08, 09, 10
3-ethylhexane	619-99-8	12.7841	0.7608	10
3-amino-1-propanol	156-87-6	12.8101	0.3141	3
acetic acid	64-19-7	12.8942	2.3249	02, 03, 06, 08, 09, 10, 11
1-hexanol	111-27-3	14.1359	1.2582	10
4-ethyl-1-octyn-3-ol	3391-86-4	14.754	1.1232	10
chloroiodomethane	75-11-6	16.3185	1.343	08, 10
2-butyl-1-octanol	3913-02-8	16.9823	1.1027	10
dimethyl disulfide	110-81-6	17.3408	1.2392	10
toluene	108-88-3	17.3869	0.9445	02, 09, 10
4-methyl-2-pentanol	108-11-2	19.1816	0.6273	3
3-hexanone	589-38-8	20.0887	0.6816	02, 03, 06, 08,
ethylbenzene	100-41-4	20.3438	0.8728	10
2-hexanone	591-78-6	20.3538	0.7042	02, 03, 06, 08
d_5_-chlorobenzene (nternal standard)	3114-55-4	20.8198	1.0564	N/A
4-hydroxy-4-methyl-2-pentanone	123-42-2	21.9885	1.1426	10
phenylethyne	536-74-3	22.0986	0.8481	03, 06,
2-(diethylamino)acetonitrile	926-64-7	22.1427	0.878	02, 08
*p*-xylene	106-42-3	22.7226	1.0333	10, 11
styrene	100-42-5	22.7794	0.8974	03, 06, 10
2-heptanone	110-43-0	23.3662	1.2113	10
1,2,4-trimethylbenzene	89-05-4	23.8049	0.7724	10, 11
1-ethyl-4-methyl-benzene	40307-11-7	24.0504	1.1884	10
bromobenzene	108-86-1	24.4771	1.127	10
2,5-hexanedione	110-13-4	24.6884	0.9548	02, 08
propyl-benzene	103-65-1	25.2494	1.1007	10
1-ethyl-2-methylbenzene	611-14-3	25.6803	2.253	10
3,4-difluorobenzaldehyde	34036-07-2	25.7777	1.0323	9
*α*-methylstyrene	98-83-9	25.7793	0.8087	02, 08, 09
benzaldehyde	100-52-7	26.3162	1.3763	02, 03, 06, 08, 09, 10
4-hydroxybutanoic acid	156-54-7	26.3403	1.452	9
1,3,5-trimethylbenzene	108-67-8	27.6137	1.4863	10
benzonitrile	100-47-0	27.6444	1.3251	02, 06, 08, 09
2-ethyl-1-hexanol	104-76-7	27.7621	1.0051	10
benzophenone	119-61-9	27.7682	1.0725	3
benzofuran	271-89-6	27.7718	0.837	02, 03, 06, 08, 09
phenol	108-95-2	28.4681	0.7994	02, 03, 06, 09
2-chlorocyclohexanol	1561-86-0	28.8662	1.3284	10
heptan-2-amine	123-82-0	29.6069	0.9578	8
1-ethyl-2,3-dimethylbenzene	933-98-2	29.9339	1.2202	10
acetophenone	98-86-2	30.2219	1.1558	02, 03, 06, 08, 09
methenamine	100-97-0	30.3213	0.8259	09, 10
benzoic acid methyl ester	99-94-5	30.5664	1.1904	02, 03, 06, 08, 09, 10, 11
*α*,*α*-dimethylbenzenemethanol	617-94-7	30.6917	1.4741	9
dibenzofuran	132-64-9	43.4907	1.1875	02, 03

**Table 3 molecules-26-03726-t003:** Heat map of variance analysis conducted on the 22 compounds that met the F_crit_ value demonstrating variance testing across all subjects. Compounds that surpassed the F_crit_ value are labelled in green (■), compounds detected in subjects’ breath, but did not surpass the F_crit_ value are labelled in yellow (■), and compounds that were not detected in a subjects’ breath samples are labelled in red (■). Compounds are listed in order from top to bottom based on the rate of identification across the trial.

Compounds	02	03	06	08	09	10	11
acetic acid							
dimethoxymethane							
benzoic acid methyl ester							
*n*-hexane							
benzaldehyde							
benzene							
benzofuran							
acetophenone							
benzonitrile							
phenol							
2-hexanone							
3-hexanone							
styrene							
iodomethane							
methylene chloride							
chloroiodomethane							
3-methyl-2-butanamine							
α,α-dimethylbenzenemethanol							
1,2,4-trifluorobenzene							
3-ethylhexane							
2,4-dimethylhexane							

## Data Availability

The data presented in this study are available on request from the corresponding author. The data are not publicly available due to privacy restrictions.

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
