# Peer review of "Pilot Study on Exhaled Breath Analysis for a Healthy Adult Population in Hawaii"

_molecules, 2021, doi:10.3390/molecules26123726_

Round 1
Reviewer 1 Report
This manuscript reports a pilot study on exhaled breath analysis using comprehensive two-dimensional gas chromatography for a healthy adult population in Hawaii.
Well established state-of-the-art methods including exhaled breath sampling on thermal desorption tubes, multi-dimensional gas chromatography and Fisher Ratio analysis were applied to a pilot trial of 7 subjects.
In general, I think further experiments are required for valid data interpretation to investigate the variance of breath compounds in a Hawaiian population.
In my opinion, major revisions are required before this article can be accepted for publication.
- Introduction:
- Overall, the structure of the introduction is comprehensible and gives the reader a good background of the importance of exhaled breath analysis. However, information such as the benefit of using GCxGC and dual detection by FID and qMS can improve readability. In addition, few references were made to GCxGC applied to exhaled breath.
- The manuscript states: ‘The number of subjects was kept intentionally small for this first pilot study in an attempt to look at intra-individual differences over time, from data collection on three separate visits for each individual.’ This approach is reasonable, since huge variations can occur from e.g. exogenous sources such as the intake of food. I was wondering if the authors could provide data and discussion on the intra-individual differences over time.
- Results:
- 1. Pre-Trial Tube Selection
- The selection of the three tube types described is appropriate.
- I was wondering if adsorbent artifact formation has been taken into account for the comparison of the total peak areas.
- The liquid injection of reference compounds with subsequent thermal desorption allows not for the verification of a suitable sampling volume, since breakthrough volumes and influencing factors such as the high humidity of the exhaled breath are not taken into account. Would the authors mind to clarify how using the reference mix verifies adequate sampling volume? I was wondering if the authors checked for moisture effects and breakthrough volumes.
- Usually, different adsorbent materials required different desorption parameters for an optimized analysis. Have the desorption parameters been optimized for the different thermal desorption tubes?
- Regarding the statement, ‘Tenax TA generally appeared to have a higher abundance of compounds for each individual’, I wondered if the authors would provide data for the other two subjects 1 and 3 as well.
- It would be interesting if the statements made in this section with regard to variability and abundance are also applicable to the exhaled breath compounds verified in section 2.2. Can the authors comment on that, please?
- 2. Post-Trial Compound Identification:
- ‘Peaks are broader in the GC×GC-qMS plot than in the GC×GC-FID plot because of differences in the detector scan rate’. Differences in the acquisition frequencies should have no influence on the peak widths of the second dimension. Can the authors clarify that, please? Are there any data available using GCxGC-FID with different acquisition frequencies to verify the statement? Reasons for peak broadening are more likely due to e.g. flow differences.
- 164-166: ‘Based on the presence of peaks in the FID and the matching MS identification, the total number of compounds that could be tentatively identified in should in Table 2.’ Please check the grammar.
- 171-173: ‘Upon collection of samples, breath samples and control samples were gathered directly after one another to reduce the sample composition variability.’ No information on control samples can be find in section 4. Materials and Methods. Can the authors please provide some information on control samples?
- Table 1: Except for subject 10, the total number of peaks detected by GCxGC-FID ranged from 20 to 30, with the use of GCxGC appearing to be excessive. The authors state ‘Although ultimately, the number of compounds of focus per subject was few, the compounds of importance were different from subject to subject, demonstrating that an analytical technique that can theoretically resolve every possible compound of importance from the potential hundreds of compounds that can appear in a contour plot is beneficial. The compounds of importance identified in the feature reduction process would not necessarily have been the same compounds highlighted in a one-dimensional GC approach that does not sufficiently resolve all possible components for further processing and characterization.’ Can the author please prove the statement with facts, e.g. by comparing the one-dimensional data? GCxGC is a complex, but powerful technique with many advantages. Unfortunately, this cannot be inferred from the data displayed. Coelutions by one-dimensional gas chromatography could also be resolved by deconvolution of mass spectra. The authors should comment on why this was not used.
- Table 2: Two of the most commonly reported exhaled VOCs in the literature are acetone and isoprene, which were not detected in the study. Can the authors comment on that, please?
- Usually, differences in breath profiles related to metabolic disorders are reflected in up- or down-regulated VOC concentrations. I wondered if differences in the concentration ranges of specific compounds were found for different subjects.
- Since a study with only 7 subjects has limited statistical validity, further experiments are recommended for conclusions on the baseline breath profile of a healthy Hawaiian population.
- 1. Pre-Trial Tube Selection
Author Response
This manuscript reports a pilot study on exhaled breath analysis using comprehensive two-dimensional gas chromatography for a healthy adult population in Hawaii.
Well established state-of-the-art methods including exhaled breath sampling on thermal desorption tubes, multi-dimensional gas chromatography and Fisher Ratio analysis were applied to a pilot trial of 7 subjects.
In general, I think further experiments are required for valid data interpretation to investigate the variance of breath compounds in a Hawaiian population.
In my opinion, major revisions are required before this article can be accepted for publication.
We would like to thank the reviewer for their thorough review and detailed comments. We feel that we have addressed these comments as outlined below and we are grateful for the improved quality they have provided within the manuscript. This was an extremely valuable review and helped us to think critically of the work we presented.
Introduction:
Overall, the structure of the introduction is comprehensible and gives the reader a good background of the importance of exhaled breath analysis. However, information such as the benefit of using GCxGC and dual detection by FID and qMS can improve readability. In addition, few references were made to GCxGC applied to exhaled breath.
Thank you for commenting on the quality of the background. We have added additional information about the use of GC×GC on lines [79-89] with more recent references to the use of GC×GC in the field of exhaled breath analysis. We have also added some information on the benefits of dual detection by qMS/FID within the manuscript on lines [100-106].
The manuscript states: ‘The number of subjects was kept intentionally small for this first pilot study in an attempt to look at intra-individual differences over time, from data collection on three separate visits for each individual.’ This approach is reasonable, since huge variations can occur from e.g. exogenous sources such as the intake of food. I was wondering if the authors could provide data and discussion on the intra-individual differences over time.
Thank you to the reviewer for outlining that this is a reasonable approach. Our opinion is that it was particularly important to start providing any possible data on our local population given the complete absence of data on this population and region in the current literature. The population size was originally intended to include 20 subjects, but as human subjects research is concerned, there are numerous challenges with recruiting and maintaining participants in these studies and as a result we were only able to report on a smaller initial cohort than anticipated. As part of our ethical approval and in line with all IRB regulations, we have to allow subjects the ability to withdraw at any point, but the study also has a defined timeline by granting agencies funding the work. We have tried to limit the conclusions of the study to only those that could be made based on the data presented, and not to overestimate results based on the small population initially sampled. As such, variance within the dataset was a huge focus so as to only report out the compounds which could be reasonably included based on inter- and intra- group variation. The intra-individual differences over time are important as the reviewer points out. The intraindividual differences are indeed a part of the Fisher Ratio calculation for every compound since they take into account the within class variance (variance of all samples from a subject across their visits). Therefore, if a compound significantly fluctuated at a high rate from visit-to-visit, it would have been removed from the final list of compounds included in the heat map table or be labelled red within that particular square. We have discussed the intraindividual differences further on lines [258-268] and lines [270-283] to outline how that was taken into account and exactly how the table presents that information to the reader for improved clarity.
Results:
- Pre-Trial Tube Selection
The selection of the three tube types described is appropriate.
Thank you for this comment and we appreciate this point.
I was wondering if adsorbent artifact formation has been taken into account for the comparison of the total peak areas.
Yes, this has been taken into account because the control samples would exhibit the levels of adsorbent artifacts expected. This is a good point and we have added further detail on this in lines [417-424].
The liquid injection of reference compounds with subsequent thermal desorption allows not for the verification of a suitable sampling volume, since breakthrough volumes and influencing factors such as the high humidity of the exhaled breath are not taken into account. Would the authors mind to clarify how using the reference mix verifies adequate sampling volume? I was wondering if the authors checked for moisture effects and breakthrough volumes.
This is a great point, and it was one reason why we chose to have both liquid injection of samples and real breath samples as a point of comparison for the different types of sorbents. We recognize it was not actually stated this way in the manuscript and as such we have adjusted the statement on lines [125-137]. We agree that liquid injection of samples is not sufficient solely as you don’t have all the same parameters you do with exhaled breath. The reference mix was used only as a semi-quantitative means of investigating sorbent comparisons. Each tube was also injected with the internal standard which was used to normalize peak areas so that only relative peak areas were compared across the sorbent tubes.
Usually, different adsorbent materials required different desorption parameters for an optimized analysis. Have the desorption parameters been optimized for the different thermal desorption tubes?
In consultation with the sorbent tube supplier, we discussed this and ultimately came to the conclusion that the three tubes being compared in the study had optimal desorption conditions within the same range, since they are very similar sorbents. If we were comparing sorbent beds that are used in very different applications, desorption temperature and times would surely have to be adjusted for optimal recovery. We used the desorption parameters from a recent tube comparison study for in vivo and in vitro sample collection using the same sorbent tubes we used (and a variety of other similar ones) by Franchina et al. (2021, Talanta) which is referenced in the paper. We have added a statement about the comparability of desorption parameters on lines [171-172]
Regarding the statement, ‘Tenax TA generally appeared to have a higher abundance of compounds for each individual’, I wondered if the authors would provide data for the other two subjects 1 and 3 as well.
Thank you for inquiring about the additional data. We included only one subject because we felt that this accurately reflected the overall trends seen in our breath data, standard injections, and in comparison to other research. We thought it would have been challenging to place all chromatograms from all subjects within the paper. As such, we have included a supplementary figure as supporting information and added some further description within the text on lines [159-162].
It would be interesting if the statements made in this section with regard to variability and abundance are also applicable to the exhaled breath compounds verified in section 2.2. Can the authors comment on that, please?
This is certainly a good point. The pre-trial tube selection study in 2.1 was completed using one dimensional GC. Therefore, the abundances and the variability are not necessarily comparable from one study to the other. However, resolution was significantly enhanced between compounds in the full trial, as the GC×GC provided the ability to separate compounds further. The range of compounds observed between the two trials was comparable but the variability from sample to sample was improved when using GC×GC. We have added a statement to this effect on lines [173-177].
- Post-Trial Compound Identification:
‘Peaks are broader in the GC×GC-qMS plot than in the GC×GC-FID plot because of differences in the detector scan rate’. Differences in the acquisition frequencies should have no influence on the peak widths of the second dimension. Can the authors clarify that, please? Are there any data available using GCxGC-FID with different acquisition frequencies to verify the statement? Reasons for peak broadening are more likely due to e.g. flow differences.
In fact, acquisition rate of the detector is a main determinant of peak width in the second dimension when you are functioning at the detector limit. Conventionally, GC×GC has been operated using TOFMS detectors which can scan at > 100 Hz. In conventional 1D GC, the detector has to be capable of acquiring spectra at a slower rate (e.g. 5-10 Hz) in order to collect the necessary minimum of 10 points across the one-dimensional peak. In GC×GC, the one-dimensional peak is sliced into multiple pieces by the modulator resulting in peaks widths much narrower (e.g. 200-300 ms wide). In order to get 10 points across the narrower peak, your detector has to function at around 50 Hz or higher. This is the generally accepted value. In our applications, we use a cheaper mass spectrometer (the qMS rather than TOFMS) to collect spectra at the max acquisition rate of 41.5 Hz, which means that we typically see broader peaks as the detector is acquiring more slowly and taking fewer spectra across an analyte peak (e.g. widening the base). However, with the dual detection system, we are acquiring on the FID at 120 Hz. This is why we use the qMS data for identifying peaks, since the chromatographic peak shape is not as sharp. The FID data are used for quantitative purposes and any statistical analyses. This approach was mentioned in lines [199-202] but we recognize that it needs some further clarification within the methods. This has now been outlined in the manuscript on lines [550-559] and references to prior work that describes the dual detection approach and acquisition data are provided. Since dual detection was not the purpose of this study we prefer to provide reference to other sources.
164-166: ‘Based on the presence of peaks in the FID and the matching MS identification, the total number of compounds that could be tentatively identified in should in Table 2.’ Please check the grammar.
Thank you for pointing this typo out. The text was adjusted to read “Based on the presence of peaks in the FID and the matching MS identification, the total number of compounds that could be tentatively identified are shown in Table 2”.
171-173: ‘Upon collection of samples, breath samples and control samples were gathered directly after one another to reduce the sample composition variability.’ No information on control samples can be find in section 4. Materials and Methods. Can the authors please provide some information on control samples?
This was an error on our part for not including and we apologize for not including it in the original manuscript. A description of control sample collection has been added to lines [417-424].
Table 1: Except for subject 10, the total number of peaks detected by GCxGC-FID ranged from 20 to 30, with the use of GCxGC appearing to be excessive. The authors state ‘Although ultimately, the number of compounds of focus per subject was few, the compounds of importance were different from subject to subject, demonstrating that an analytical technique that can theoretically resolve every possible compound of importance from the potential hundreds of compounds that can appear in a contour plot is beneficial. The compounds of importance identified in the feature reduction process would not necessarily have been the same compounds highlighted in a one-dimensional GC approach that does not sufficiently resolve all possible components for further processing and characterization.’ Can the author please prove the statement with facts, e.g. by comparing the one-dimensional data? GCxGC is a complex, but powerful technique with many advantages. Unfortunately, this cannot be inferred from the data displayed. Coelutions by one-dimensional gas chromatography could also be resolved by deconvolution of mass spectra. The authors should comment on why this was not used.
We appreciate the inquiry about the necessity of using GC×GC. In fact, we deemed GC×GC to be unnecessary in assessing the sorbent tube comparison, since we were mostly concerned about range of analytes and total peak area recovered, and this was the justification for not using GC×GC for that part of the research. However, the focus of this paper was not to provide a comparison of one-dimensional vs. two-dimensional data, as this has been done repeatedly in the literature for VOC analysis and was outside the scope of this research. Frankly these types of papers that simply directly compare GC with GC×GC data are considered uninteresting to journals as the power of GC×GC for complex sample analysis such as metabolomics has been already established in the separation science community.
The data presented in Table 1 is preceded by a paragraph that explains this in more detail on lines [199-219]. Here it is stated that although there is a lower number of compounds that exceed Fcrit (22) the actual number of compounds detected OVERALL in the trial surpassed 100. Though the actual number of compounds in Table 1 appears to be low, the number of peaks physically detected in the total trial warrants the use of GC×GC and allows for sufficient separation of those compounds. The numbers reported are based on the statistical analysis applied, and not based on the number of raw component peaks in the chromatogram. This is extremely important when performing an untargeted analysis. We recognize that this point may not have come across in the original text and that it sounds like only 20-30 compounds were separated in the study. We only presented the compounds that were (1) not representative of analytical artifacts, (2) present at levels that were variable (up- or down-regulated) compared to the control samples, and (3) had compound identifications that were reliable enough to report the identity of the compound. We have adjusted the text to explain these points on lines [208-216]. We also hope that the further outlining of the use of GC×GC added to the introduction helps to justify the choice of analytical technique for the research.
Table 2: Two of the most commonly reported exhaled VOCs in the literature are acetone and isoprene, which were not detected in the study. Can the authors comment on that, please?
This is a good and relevant point. It should be noted that the samples collected in this study were done so onto sorbent tubes. Tenax TA compound range begins at C6 and both acetone and isoprene are small molecules with very low boiling points (isoprene especially). To collect these molecules onto sorbent based techniques is much more challenging and this is probably one reason why they are not reported herein. We injected standards of isoprene and acetone onto sorbent tubes early on in our research and we had a challenging time to detect them. When injected as liquid samples we are able to see these compounds early in analysis but using the sorbent tubes and thermal desorption via a cold trap they are harder to capture. As such, we did a mining of the literature and found that most studies monitoring these compounds do so by direct analysis techniques such as SIFT-MS where the sample can be injected directly and there is no potential loss of these very volatile analytes. We agree this is important to point out and we have added text within the paper on lines [334-340] to explain the absence of these compounds and suggest how they could be investigated with further work.
Usually, differences in breath profiles related to metabolic disorders are reflected in up- or down-regulated VOC concentrations. I wondered if differences in the concentration ranges of specific compounds were found for different subjects.
The concentration ranges were different from subject to subject. However, as we were not monitoring any metabolic disorders we did not report these absolute concentrations directly. We performed quantitation only relative to an internal standard and not in a manner that allowed absolute concentrations to be directly investigated, so all magnitudes were relative to one another. This is typical of non-targeted discovery work and we agree that this could be a step for further work when a specific metabolic disorder marker is being investigated, to perform full quantification and provide those magnitude differences. In performing this pilot study, the first step was to identify compounds of interest. In a larger study with more individuals you would be better able to make inferences in concentration differences from person to person, and this should be a focus moving forward with this work. We have added this to the discussion on lines [341-346].
Since a study with only 7 subjects has limited statistical validity, further experiments are recommended for conclusions on the baseline breath profile of a healthy Hawaiian population.
We agree and this is included in the manuscript on lines [349-350] and [356-357].
Reviewer 2 Report
1, Figure 1
Fig 1 said: “Figure 1. Total peak area obtained using…”. The plot is “relative peak area”. Is it relative or total peak area? If it is “relative peak area, it is relative to what? This is not clear.
2, Figure 2
The results in figure are misleading. Based on the sample collection method, breath air was collected in Bio-VOC sampler, and then transferred into cartridges immediately. That means that samples loaded onto three different types of cartridges were “different samples from the same participant”. You cannot assume different samples from the same participant are the same. Results in Fig. 2 include the differences between cartridges and also the difference between the samples.
3, line 338-339, “TD tube was reconditioned for 30 min at 300 °C with a flow of ultra high purity nitrogen (Airgas) with a pressure of 20 psi.”
What was the reconditioning flow rate? For cartridge condition, conditioning flow rate is the deciding factor, not the pressure.
4, Line 383-385, “The GC oven started at an initial temperature of 60 °C, was increased to a final temperature 250 °C at a rate of 3 °C/min, and held for 5 min, resulting in a total run time of 46 min.”
This is absolutely not a 46min run. (250-60)/3+5=68.33min. The plots in Figure 3 were 46min. Then what were the correct GC conditions?
5, Line 400-403
The thermal desorption unit has an extra N2 supply for desorption? Not using He for the GCMS system?
Author Response
- Figure 1
Fig 1 said: “Figure 1. Total peak area obtained using…”. The plot is “relative peak area”. Is it relative or total peak area? If it is “relative peak area, it is relative to what? This is not clear.
Thank you for pointing out this ambiguity in the figure heading. The data is normalized to an internal standard and then a sum was taken for all relative peak areas. We adjusted the figure heading to read “sum of relative peak area for all normalized compounds of interest” rather than “total peak area” on line [141].
- Figure 2
The results in figure are misleading. Based on the sample collection method, breath air was collected in Bio-VOC sampler, and then transferred into cartridges immediately. That means that samples loaded onto three different types of cartridges were “different samples from the same participant”. You cannot assume different samples from the same participant are the same. Results in Fig. 2 include the differences between cartridges and also the difference between the samples.
This is a very valuable point that we did not think to include in the text of the manuscript but will assist readers in their interpretation. We have included a point about this directly in the figure heading to help make this point on lines [151-153], included plots of Subject 1 and 3 in the supplementary information so that trends can be viewed across additional subjects, and also added a point on this in the text on lines [159-162].
- line 338-339, “TD tube was reconditioned for 30 min at 300 °C with a flow of ultra high purity nitrogen (Airgas) with a pressure of 20 psi.”
What was the reconditioning flow rate? For cartridge condition, conditioning flow rate is the deciding factor, not the pressure.
We used an external instrument, the TC-20 (tube conditioner) to condition tubes. This was not directly stated previously and has been added to the manuscript. On the tube conditioner, you only control flow by setting pressure, therefore we only included the pressure setting. There is no flow controller on this instrument and you adjust the pressure to a level based on attaching a flowmeter to the outlet of the tubes. We have added the equivalent flow rate to the pressure that comes directly from the manual and was used in calibrating the pressure and flow of this instrument.
- Line 383-385, “The GC oven started at an initial temperature of 60 °C, was increased to a final temperature 250 °C at a rate of 3 °C/min, and held for 5 min, resulting in a total run time of 46 min.”
This is absolutely not a 46min run. (250-60)/3+5=68.33min. The plots in Figure 3 were 46min. Then what were the correct GC conditions?
You are correct and thank you for catching this mistake on our part. This was a typo when we input the values into the description of the method, not an error in the figure. The oven program was updated in the text to read “The GC oven started at an initial temperature of 60 °C, held for 3 min, was increased to a final temperature 250 °C at a rate of 5 °C/min, and held for 5 min, resulting in a total run time of 46 min” which is written on lines [480-482].
- Line 400-403
The thermal desorption unit has an extra N2 supply for desorption? Not using He for the GCMS system?
This point was clarified in the prior description of the TC-20. UHP Nitrogen is the recommended gas for the TC-20 conditioning system. We also added an extra explicit statement on the use of Helium for carrier gas for both systems. This was included previously for the GC-MS system but not for the GC×GC-qMS/FID system. It is now on line [478].
Reviewer 3 Report
The authors present a preliminary study designed to explore baseline data on respiratory gas profiles in the healthy Hawaiian population. Analytically, they propose the use of GCxGC and show results of a small cohort study.
The objective and scope of the manuscript (generating and presenting the baseline level for BGA) could be interesting for the readers of Molecules as well as for breath gas analytics in general. However, the quality of the data presented as well as the general quality of the paper must be strongly improved prior to be suitable for the publication in a journal. Beside to several small mistakes, probably due to a fast proofreading, important information in support of the data presented are missing. The research work, as it is described in the submitted version, seems to be in a very preliminary stage and large parts of the discussion and results are very superficial and do not lead to new results/findings. It is more an internal test of the equipment than an applications study. Both objectives of the manuscript, namely the application of GC×GC as well as proposing a study design for large cohort study are not supported by the results. The quality (and quantity) of large parts of the results is far behind what can be found in the literature and what has been cited (in parts).
Title: The title indicates a focus on GC×GC. However, results do not show new findings. The abstract and discussion does more emphasize the cohort study to provide a baseline profile for respiratory gas.
Introduction:
A more comprehensive and critical overview of the application of GC(×GC) in Breath gas studies should be given. Other techniques should be also mentioned in relation to a cohort study.
Results/Discussion:
The results part does not provide new findings or approval/disapproval for published but still not yet generally accepted results. One reason for this is the rather superficial evaluation and discussion of the own data. E.g. Fig. 1 shows total peak areas, which is a very (very) rough indicator. Variations for individual compounds or basic statistics (significant) are not given. In general it is not clear why, 3 different GC(xGC) techniques were applied. No data for the added value is shown. All findings are already published in other much more extensive studies. Same holds for figure 3. What are the new findings or results here?. I also doubt that the peak broadening for the MS data is solely due to the acquisition frequency. In addition, the geometry of the ionization interface of the MS might be a reason.
Table 2 gives only raw data. The selection criteria is not explained (match criteria of NIST, RI?). Raw retention times are not very helpful in GC(xGC) in order to reproduce the method and compare results.
Table 3: How did you calculate/ determine F(crit)?
Method Part:
The Method Part is very filthy and seems to be written in a hurry. Very basic proof reading seems to be not yet complete. (E.g. Line 436, 448 only as two examples). Much information is not provided or should be revised. E.g. Total run times of the method can not be derived from the given parameters.
The application of liquid sample material on absorption material is not suitable to derive any conclusions for it’s behavior for BGA.
In summary, I would not recommend the authors to conduct a larger cohort study based on the results shown.
Author Response
The authors present a preliminary study designed to explore baseline data on respiratory gas profiles in the healthy Hawaiian population. Analytically, they propose the use of GCxGC and show results of a small cohort study.
The objective and scope of the manuscript (generating and presenting the baseline level for BGA) could be interesting for the readers of Molecules as well as for breath gas analytics in general. However, the quality of the data presented as well as the general quality of the paper must be strongly improved prior to be suitable for the publication in a journal. Beside to several small mistakes, probably due to a fast proofreading, important information in support of the data presented are missing. The research work, as it is described in the submitted version, seems to be in a very preliminary stage and large parts of the discussion and results are very superficial and do not lead to new results/findings. It is more an internal test of the equipment than an applications study. Both objectives of the manuscript, namely the application of GC×GC as well as proposing a study design for large cohort study are not supported by the results. The quality (and quantity) of large parts of the results is far behind what can be found in the literature and what has been cited (in parts).
Title: The title indicates a focus on GC×GC. However, results do not show new findings. The abstract and discussion does more emphasize the cohort study to provide a baseline profile for respiratory gas.
We have removed GC×GC from the title of the study as we agree it was not the main point of this research. While we provided data to justify the use of GC×GC for those who are not familiar with the technique, it should not be the main point of the article.
Introduction:
A more comprehensive and critical overview of the application of GC(×GC) in Breath gas studies should be given. Other techniques should be also mentioned in relation to a cohort study.
At the request of Reviewer 1 and Reviewer 3 we added more references and description of the overview of GC×GC for background.
Results/Discussion:
The results part does not provide new findings or approval/disapproval for published but still not yet generally accepted results. One reason for this is the rather superficial evaluation and discussion of the own data. E.g. Fig. 1 shows total peak areas, which is a very (very) rough indicator.
We agree this is only a single indicator, which is why it was supported with chromatograms. We also added additional figures in the supporting information as well in this round of review. The main point within the text is that all three sorbent options are valid based on manufacturer recommendations and literature reviews that are quoted within. The data was simply added to verify what is already previously known and to continue moving forward.
Variations for individual compounds or basic statistics (significant) are not given. In general it is not clear why, 3 different GC(xGC) techniques were applied. No data for the added value is shown. All findings are already published in other much more extensive studies. Same holds for figure 3. What are the new findings or results here?. I also doubt that the peak broadening for the MS data is solely due to the acquisition frequency. In addition, the geometry of the ionization interface of the MS might be a reason.
Basic statistics on variance were demonstrated in lines [565-577] and the concept of using Fisher Ratio analysis to determine compound inclusion (i.e. significance) is discussed throughout the paper. This is a generally accepted method for establishing VOC significance in many studies using GC approaches.
The purpose of Figure 3 is to demonstrate the cumulative nature of compounds across a study that includes (as stated) only 7 subjects. In a much larger study, the number of peaks identified would be vastly increased. We added additional text on lines [208-216] about these points and also at the request of another reviewer to strengthen the presented figure.
The new results or findings are those based on a focus on a new population that has never been studied. This is stated throughout the manuscript several times. While the number of individuals is low as a starting point, there are key limiting factors in initiating this type of research within the Pacific Rim. All IRB approval require that subjects are allowed to withdraw from the study at any point in time. Recruitment is more challenging within Pacific Island populations due to cultural norms, and to date there has been no focus on this region of the world for exhaled breath analysis. We originally targeted 20 individuals and this is what our funding was provided for. Ultimately, only 3 individuals in the first part of the study and 7 individuals in the second part of the study were retained. We agree this is a challenge however does not invalidate the data which was collected and the individuals who agreed to participate in the study. We maintain that providing any data on a previously unmonitored population is essential to improving our ability to continue studies like this in the future, and for improving attention around increased funding of minority population research. To simply throw this data away and not publish it because it is a small dataset would be very unfortunate, given that no other previous research has taken this approach. We targeted a journal that we felt was appropriate for the nature of this study, and did not target a medical journal as we agree that there are key limitations that prevent this data from being standalone without further input in the field. It is our hope that by publishing this work, we will be able to draw attention to this area of the world and be able to attract more interest in larger studies. While we agree that the data set is small, we respectfully disagree that no new findings are presented. The limitations of the study have been discussed in detail within the discussion section and clearly states the more data must be collected, on larger populations, with more robust statistics, for any broad inferences to be made.
Table 2 gives only raw data. The selection criteria is not explained (match criteria of NIST, RI?). Raw retention times are not very helpful in GC(xGC) in order to reproduce the method and compare results.
We believe a list of compounds and their location on the GC×GC space is helpful for future work. Raw retention times give users an idea of elution order and the structure of the chromatogram for comparison. This is particularly important since a non-standard column set is used. In fact, another researcher in the field of exhaled breath analysis recently reached out to us to get a list of retention time for compounds on our specific columns, so we think this is valuable to the community. This table also includes which specific subject each compound was found in and therefore we believe this type of data is helpful. Without this table there would be no list of compounds provided and this makes future work comparing to this study challenging.
Selection criteria are a matter of methods (not for a table header) and were included in that section. All selection factors are listed in the methods in section 4.7 on lines [512-540]. Beyond chromatographic data selection factors, the FR approach is further detailed for selection on lines [571-577].
Table 3: How did you calculate/ determine F(crit)?
The Fcrit value is described in the materials and method section on line [566]. The Fcrit determination is coming from the number of groups of samples, the number of samples present in each group, and the significance level of 0.05. This gets adjusted based on the study design. It is similar to looking up a
Method Part:
The Method Part is very filthy and seems to be written in a hurry. Very basic proof reading seems to be not yet complete. (E.g. Line 436, 448 only as two examples). Much information is not provided or should be revised. E.g. Total run times of the method can not be derived from the given parameters.
We have performed another careful proofreading and we are not sure what is specifically meant by the words “very filthy”. Line 436 was a residual in-text comment we had forgotten to remove – thank you for that. Line 448 is accurate based on software manufacturer recommendation, so we are not sure what has been recommended here. It is our belief that any “settable” parameter should be stated and written out, so this is what we have provided. While the methods are lengthy, they include any parameter needed for reproducing the method. Unfortunately, many published untargeted studies do not report this extensive parameter list which makes reproducibility challenging. The total run time issue was clarified as outlined in the reviewer 2 comments, as this was a typo on our part. All run times should be able to be calculated based on the oven methods provided and have also been stated in text, and can further be verified by the chromatograms provided as well. The reviewer checked off “I do not feel qualified to judge about the English language and style” on the report form. We have addressed proofreading in the revisions and thank the reviewer for a reminder to go through in further detail.
The application of liquid sample material on absorption material is not suitable to derive any conclusions for it’s behavior for BGA.
This was also addressed in response to reviewer 1. It was one aspect of the verification we performed. Results of liquid standards, individual breath samples, literature comparisons, and manufacturers recommendations were all provided. It is one small piece of the full information. We recognize that this would not be suitable alone and it is why we did not perform it alone. Also it is mentioned in the text that all 3 sorbent tubes are choices that are considered valid within the literature, and that there are only subtle differences between the three sorbents. This is not new information, we were simply providing some in house data to justify the choice which was made.
In summary, I would not recommend the authors to conduct a larger cohort study based on the results shown.
We thank the reviewer for this input and we will take this under advisement. We do hope that further studies will focus on this particular population so that breath analysis can hopefully become a tool used in areas such as Hawaii and the Pacific Islands where there are severe health disparities for many of the current applications of exhaled breath diagnostics – e.g. alarming rates of COPD that have only been studied in other parts of the world on subjects with very different racial backgrounds. Even if our results from this study are not the foundation for future work, we believe that making exhaled breath analytics accessible to different populations around the world should be a focus of developing research, rather than continuing to study similar demographics that have been more extensively focused on in research.
Round 2
Reviewer 1 Report
Thank you very much for the clarifications. Most of the questions asked have been answered and supporting information has been added, improving the overall manuscript.
One point has not yet been clarified that should be revised.
2. Post-Trial Compound Identification:
‘Peaks are broader in the GC×GC-qMS plot than in the GC×GC-FID plot because of differences in the detector scan rate’. Differences in the acquisition frequencies should have no influence on the peak widths of the second dimension. Can the authors clarify that, please? Are there any data available using GCxGC-FID with different acquisition frequencies to verify the statement? Reasons for peak broadening are more likely due to e.g. flow differences.
In fact, acquisition rate of the detector is a main determinant of peak width in the second dimension when you are functioning at the detector limit. Conventionally, GC×GC has been operated using TOFMS detectors which can scan at > 100 Hz. In conventional 1D GC, the detector has to be capable of acquiring spectra at a slower rate (e.g. 5-10 Hz) in order to collect the necessary minimum of 10 points across the one-dimensional peak. In GC×GC, the one-dimensional peak is sliced into multiple pieces by the modulator resulting in peaks widths much narrower (e.g. 200-300 ms wide). In order to get 10 points across the narrower peak, your detector has to function at around 50 Hz or higher. This is the generally accepted value. In our applications, we use a cheaper mass spectrometer (the qMS rather than TOFMS) to collect spectra at the max acquisition rate of 41.5 Hz, which means that we typically see broader peaks as the detector is acquiring more slowly and taking fewer spectra across an analyte peak (e.g. widening the base). However, with the dual detection system, we are acquiring on the FID at 120 Hz. This is why we use the qMS data for identifying peaks, since the chromatographic peak shape is not as sharp. The FID data are used for quantitative purposes and any statistical analyses. This approach was mentioned in lines [199-202] but we recognize that it needs some further clarification within the methods. This has now been outlined in the manuscript on lines [550-559] and references to prior work that describes the dual detection approach and acquisition data are provided. Since dual detection was not the purpose of this study we prefer to provide reference to other sources.
I agree that a slower aquisition frequencie can be used for compound verification applying GCxGC-qMS. However, the explanation for the peak broadening (following statement) is unsatisfactory because of the lack of facts.
'Peaks are broader in the GC×GC-qMS plot than in the GC×GC-FID plot because of differences in the detector scan rate. The GC×GC-FID scan rate is much faster than the GC×GC- qMS scan rate, meaning that some peak broadening can occur in the second dimension for narrow GC×GC peaks.'
Slowing down the acquisition frequency can result in an undersampling of the peak. The software would then adapt a Gaussian profile, which can lead to a broader peak. The peak width would be artificially wider. If you look into literature, there are some interesting studies showing the influence of the acquisition frequency. I would recommend having a look at 'Evaluation of a Rapid-Scanning Quadrupole Mass Spectrometer in an Apolar × Ionic-Liquid Comprehensive Two-Dimensional Gas Chromatography System by Purcaro et al. (Anal. Chem. 2010, 82, 8583–8590)' and 'Considerations on the determination of the limit of detection and the limit of quantification in one-dimensional and comprehensive two-dimensional gas chromatography by Krupčík et al. (Journal of Chromatography A, Volume 1396, 29 May 2015, Pages 117-130)'. In these studies, changing the aquisition frequencies clearly shows no such affect on the peak widths, as state by the authors. Therefore, the explanation for the broader peaks is still not justified by lower aquisition frecencies. However, it is more lokely that reasons for peak broadening are e.g. flow differences, which is also suported by the differences in retention times of the second dimension, as can be seen from the contour plot.
I would recommend changing or removing the following statement.
'Peaks are broader in the GC×GC-qMS plot than in the GC×GC-FID plot because of differences in the detector scan rate. The GC×GC-FID scan rate is much faster than the GC×GC- qMS scan rate, meaning that some peak broadening can occur in the second dimension for narrow GC×GC peaks.'
Author Response
Manuscript ID: molecules-1202253
Title: Pilot study on exhaled breath analysis using comprehensive two-dimensional gas chromatography for a healthy adult population in Hawaii
Please note: Line numbers in response to reviewer comments refer to the line numbers in the track changes version of the manuscript, with “all markup” view selected.
Reviewer 1
Thank you very much for the clarifications. Most of the questions asked have been answered and supporting information has been added, improving the overall manuscript.
One point has not yet been clarified that should be revised.
I agree that a slower aquisition frequencie can be used for compound verification applying GCxGC-qMS. However, the explanation for the peak broadening (following statement) is unsatisfactory because of the lack of facts.
Slowing down the acquisition frequency can result in an undersampling of the peak. The software would then adapt a Gaussian profile, which can lead to a broader peak. The peak width would be artificially wider. If you look into literature, there are some interesting studies showing the influence of the acquisition frequency. I would recommend having a look at 'Evaluation of a Rapid-Scanning Quadrupole Mass Spectrometer in an Apolar × Ionic-Liquid Comprehensive Two-Dimensional Gas Chromatography System by Purcaro et al. (Anal. Chem. 2010, 82, 8583–8590)' and 'Considerations on the determination of the limit of detection and the limit of quantification in one-dimensional and comprehensive two-dimensional gas chromatography by Krupčík et al. (Journal of Chromatography A, Volume 1396, 29 May 2015, Pages 117-130)'. In these studies, changing the aquisition frequencies clearly shows no such affect on the peak widths, as state by the authors. Therefore, the explanation for the broader peaks is still not justified by lower aquisition frecencies. However, it is more lokely that reasons for peak broadening are e.g. flow differences, which is also suported by the differences in retention times of the second dimension, as can be seen from the contour plot.
I would recommend changing or removing the following statement.
'Peaks are broader in the GC×GC-qMS plot than in the GC×GC-FID plot because of differences in the detector scan rate. The GC×GC-FID scan rate is much faster than the GC×GC- qMS scan rate, meaning that some peak broadening can occur in the second dimension for narrow GC×GC peaks.'
Thank you to the reviewer for taking the time to review our responses and the manuscript a second time. We greatly appreciate the dialog on this topic. As a result of the comments we have removed the statement that the reviewer has brought concern about due to the points brought forward.